# Recent Progress in Research on Ferromagnetic Rhenium Disulfide

**DOI:** 10.3390/nano12193451

**Published:** 2022-10-02

**Authors:** Hongtao Ren, Gang Xiang

**Affiliations:** 1School of Materials Science and Engineering, Liaocheng University, Hunan Road No. 1, Liaocheng 252000, China; 2College of Physics, Sichuan University, Wangjiang Road No. 29, Chengdu 610064, China

**Keywords:** defects, doping, strain, phase, domain

## Abstract

Since long-range magnetic ordering was observed in pristine Cr_2_Ge_2_Te_6_ and monolayer CrCl_3_, two-dimensional (2D) magnetic materials have gradually become an emerging field of interest. However, it is challenging to induce and modulate magnetism in non-magnetic (NM) materials such as rhenium disulfide (ReS_2_). Theoretical research shows that defects, doping, strain, particular phase, and domain engineering may facilitate the creation of magnetic ordering in the ReS_2_ system. These predictions have, to a large extent, stimulated experimental efforts in the field. Herein, we summarize the recent progress on ferromagnetism (FM) in ReS_2_. We compare the proposed methods to introduce and modulate magnetism in ReS_2_, some of which have made great experimental breakthroughs. Experimentally, only a few ReS_2_ materials exhibit room-temperature long-range ferromagnetic order. In addition, the superexchange interaction may cause weak ferromagnetic coupling between neighboring trimers. We also present a few potential research directions for the future, and we finally conclude that a deep and thorough understanding of the origin of FM with and without strain is very important for the development of basic research and practical applications.

## 1. Introduction

In 2017, Zhang et al. [1] and Xu et al. [2] discovered long-range ferromagnetic ordering in both pristine Cr_2_Ge_2_Te_6_ and monolayer CrCl_3_. Two-dimensional ferromagnetism (FM) [3,4,5] has since then gradually reached an unprecedented level. However, most of the 2D crystals with FM [1,2,3,4,5,6,7,8,9,10] have been obtained using mechanical exfoliation, and long-range ferromagnetic order can only be maintained at low temperatures. Instead, the possibility of growing samples of wafer-scale size, and with room temperature ferromagnetism (RTFM), is a prerequisite for the development of spintronic devices. New experimental methods, such as molecular beam epitaxy (MBE) [7,11,12,13], therefore, have been developed to grow large-scale materials. However, this specific method needs high-vacuum conditions, which limits its wide use. Furthermore, phase engineering [14,15,16,17], doping engineering [18,19,20,21,22,23,24,25], strain engineering [4,26,27,28,29,30,31], as well as light-driven [32,33], gate-tunable [3,34], patterning-induced [35], and sodium- [36,37] or self-intercalated [13,25] and domain engineering [37] have been used to elevate the Curie temperature (T_c_). Interestingly, transition metal phosphorus trichalcogenides (MPX_3_, where M stands for a transition metal atom and X = S and Se) have also attracted much attention [38,39]. In particular, the pristine MPS_3_ exhibited antiferromagnetic (AFM) properties. When the “M” atom is replaced by other transition metal atoms, it may drive the transition from AFM to FM. Despite some progress that has been made, only a few experimental results have reported 2D materials with RTFM [12,27,32,35,36,40].

Inspired by these efforts, many groups have tried to endow intrinsic nonmagnetic materials with magnetism. Unlike other hexagonal (*H* or 2*H*) transition metal chalcogeni-des (TMDs) with high symmetry, the unique distorted (*T_d_*) structure of rhenium disulfide (ReS_2_) has a low in-plane symmetry [41], which endows it with strong in-plane anisotropic properties [42,43,44]. It is worth noting that a series of innovations have also been made in the RTFM of ReS_2_. For example, Fu et al. [18] discovered in 2018 a ferromagnetic order in an N-doped ReS_2_ system, which was obtained using a hydrothermal method. More specifically, a phase transition from FM to AFM could be realized by controlling the doping concentration and the doping sites. Furthermore, the mechanically exfoliated and distorted monolayer showed a long-range ferromagnetic order at RT [17,45]. In addition, a biaxial tensile strain was found to enhance the RTFM in the ReS_2_ web buckles. It is worth mentioning here that the Re-related vacancies play a crucial role for the RTFM of ReS_2_. Moreover, Loh et al. [37] analyzed parallel mirror twin boundaries (MTBs) in an electrochemical exfoliated ReS_2_ monolayer by using linearly polarized optical microscopy (OM), angle-resolved polarized Raman spectroscopy (ARPRS), and scanning transmission electron microscopy (STEM). The in-plane biaxial strain in the MTBs was found to enhance the magnetic moment from 0.09 to 1.94 µ_B_/supercell. In addition, vacancy defects were created in the ReS_2_ films using Ar plasma treatments, and the cations were driven into anionic sites by using thermal annealing (to form antisite defects). Interestingly, spin polarization caused by defects can also enhance the occurrence of RTFM.

Here, we will give an overview of the timeline for the occurrence of magnetism in ReS_2_. As shown in Figure 1, we compare the various strategies for regulating the RTFM, including defect engineering, doping engineering, strain engineering, phase engineering, and domain engineering. Notably, these methods have made large experimental breakthroughs since 2018. However, only a few ReS_2_ materials have exhibited RTFM. Finally, we will present a few potential research directions for the future. In short, it is very necessary to obtain a deep understanding about the origin of RTFM and strain-tunable RTFM, which will further help the development of spintronics to flourish.

## 2. Crystal Structure and Band Structure of ReS_2_

### 2.1. Crystal Structure

Monoatomic monolayers such as graphene [44,46,47] have a hexagonal crystal structure (a so-called *H* phase), as shown in Figure 2A. The graphene structure is planar, which is due to the sp^2^ hybridization of the carbon atoms. However, the occurrence of sp^3^ hybridization causes a buckled structure in, e.g., silicene and germanene.

Like monoatomic crystals, monolayer TMDs consist of three layers of atoms, in which one layer of transition metal (M) atoms is sandwiched by two layers of chalcogeni-de (X) atoms. Chalcogen layers can be stacked on top of each other either as an *H* phase (i.e., with the tetrahedral holes above the transition metal atoms), as in Figure 2A, or as a *T_c_* phase (i.e., with the octahedral holes above the transition metal atoms), as in Figure 2B. There are strong covalent bonds within each layer and weaker van der Waals (vdW) bonds in between. In the octahedral phase, *T_c_*, one of the sulfur layers has been shifted with respect to the other. Notably, ReS_2_ has a stable distorted octahedral structure, as shown in Figure 2C.

### 2.2. Band Structure

Bulk ReS_2_ is a direct band gap semiconductor with a layered structure, showing novel anisotropic properties [41,48]. As shown in Figure 3A, the crystal structure of ReS_2_ with the *T_d_* phase [41] is obviously different from that of MoS_2_ with the *H* phase [49,50]. Density functional theory (DFT) calculations show that bulk (1.35 eV) and monolayer (1.43 eV) ReS_2_ have similar band structures, both of which are direct band gap semiconductors, but their band gaps are slightly different (only 80 mV difference), as shown in Figure 3B.

Actually, the adjacent layers in ReS_2_ are only weakly coupled (~18 meV) in Figure 3C, whereas those in MoS_2_ are coupled with much higher energy (~460 meV). Interestingly, when the thickness of ReS_2_ is reduced down to a single layer [51], its electronic band structure does not exhibit a transition from an indirect to a direct bandgap, which is different from that of MoS_2_. The bandgap of bulk, trilayer, and monolayer ReS_2_ are 1.35 eV, 1.40 eV, and 1.44 eV, respectively, as shown in Figure 3D. In experiment, the electrical band structure of rhenium disulfide can be accurately described by angle-resolved photoemission spectroscopy (ARPES) measurements [52]. Although the surface Brillouin zone of ReS_2_ is almost hexagonal, its electronic structure shows significant in-plane anisotropy, resulting in unique anisotropic optical and electrical properties.

## 3. Progress in Theoretical Calculations and Experimental Studies of ReS_2_ Magnetism

### 3.1. Defect Engineering

Generally, the unintentional generation of defects is unavoidable in the growth, peeling, and transferring of single-layer crystals, which often deteriorates the properties of the materials. Meanwhile, the intentional introduction of defects may induce new properties to the materials. Therefore, defect engineering [53,54,55,56] has become an important strategy to use for the modification of material properties. Experimentally, defects are often introduced into the parent materials by means of ion irradiation [57,58,59,60,61,62,63], plasma treatment [64], thermal annealing [60,63,64,65,66,67], etc. In 2014, Peter et al. [57] studied the formation energy and stability of lattice defects in monolayer ReS_2_ by using a combination of experimental and theoretical investigations. The mechanism of defect-mediated magnetism was then revealed. Peeters et al. [49] first introduced point defects in pristine ReS_2_ by using He ion irradiation. In order to understand the formation energy and stability of these defects, they also carried out first-principle calculations. Optimized atomic structures of a distorted 1*T*-ReS_2_ monolayer were then created, as shown in Figure 4. However, the introduction of defects in these optimized structures was not found to change the semiconductor properties or drive any phase transition.

It is worth mentioning that S-related defects (*V_S_*, *V_S+S,_* and *V*_2*S*_) cannot cause magnetism, whereas Re-related defects (*V_Re_*, *V_ReS_* and *V_ReS_*_2_) can. As shown in Figure 5, the magnetization comes predominantly from p orbitals of two neighboring S atoms within the vacancy region.

Interestingly, the antisite defects, such as *S_S_**_→Re_* [57,64] and *S*_2*S*_*_→Re_* [57], bring the magnetic moment of 3 µ_B_ into the supercell, as shown in Figure 6. However, no RTFM could be detected in the experiments. *V_ReS_* and *V_ReS_*_2_, thereafter, can be created by introducing biaxial tensile strain in the ReS_2_ web buckles (which exist in multiple directions in the plane and cross each other to form some web patterns) [27,68,69].

By performing theoretical calculations, we have also found that *V_Re_*, *V_ReS,_* and *V_ReS_*_2_ can produce a magnetic moment of 1–3 µ_B_/supercell, as shown in Figure 6A. The supercell size had no obvious effect on the magnetic properties of the system with *V_Re_*, *V_S_*, *V_S+S_*, *V*_2*S*_, and *S_S-Re_*. In contrast, the total magnetic moment of the supercell with *V_ReS_* and *V_ReS_*_2_ was found to be not only related to supercell size, but also related to the phase. Notably, no matter what type of defects exist, the *T_c_* phase cannot produce a magnetic moment.

In 2022, antisite defects (e.g., *S_S-Re_*) were introduced into 2D ReS_2_ flakes using Ar plasma and thermal annealing treatment [64]. Actually, the defects in ReS_2_ nanosheets were formed by the Re atoms occupying the positions of the S atoms, and new *V_Re_* defects were simultaneously introduced (Figure 6). With an increase in plasma treatment time, the magnetic moment increased at RT in the experiment. The magnetism was enhanced up to ~20 times after the subsequent thermal annealing. The significant increase in magnetism was mainly due to the introduction of antisite defects. Similarly, antisite defects (e.g., Mo_S_2__ and S_2Mo_) have also been observed in CVD-grown MoS_2_ using STEM [72,73] analysis. Even if the theory predicts that these antisite defects could induce magnetism, no relevant magnetism has been observed in the experiment so far.

### 3.2. Doping Engineering

Doping engineering has become a common strategy for adjusting the properties of a material. In 2014, Peter et al. studied the effects of substitutional doping [70] by non-metallic and metal atoms on electrical and magnetic properties. The modulation of magnetism in the ReS_2_ material was, thereafter, studied theoretically by means of non-metallic element adsorption [19], fluorination [19,21,71], transition-metal doping [74,75], and non-magnetic metal doping [76].

#### 3.2.1. Nonmetallic Element Doped ReS_2_

Actually, doping elements, substitutional sites, supercell sizes, and the distances between the adsorbed atoms and S atoms have all shown a large effect on the magnetism of ReS_2_ in Figure 6B. More specifically, F and B were shown to have the strongest effect on the magnetic properties, whereas H, N, P, As, F, and Cl had the least effect. On the other hand, S, Se, and Te showed no effect on the magnetic properties.

In 2018, Fu et al. [18] prepared N-doped ReS_2_ nanospheres with different doping concentrations by using hydrothermal methods, as shown in Figure 7. Nitrogen doping can drive the phase transition of ReS_2_ from nonmagnetic to ferromagnetic.

More specifically, nitrogen doping with different dopant concentrations has been realized by varying the mass ratio of ammonium rhenate (NH_4_ReO_4_) and thiourea (CH_4_N_2_S). The magnetic moment did reach a value of 2.1 emu/g at 2K, as shown in Figure 7A. The inset in Figure 7A shows the non-zero coercivity, indicating the presence of a magnetic anisotropy in the ReS_2_ sample. A distinct exchange bias caused by FM-AFM coupling were also observed, as shown in Figure 7C–D. However, nitrogen doping failed to induce long-range ferromagnetic ordering in the ReS_2_ system at RT.

In order to explain the correlation between the doping concentration and the magnetism for the ReS_2_ supercell with the *T_d_* phase, the magnetic moment and charge distribution were calculated using VASP, as shown in Figure 8. The supercells with different doping concentrations had the following magnetic moments: 0.703 µ_B_ for 1N per supercell, 1.522 µ_B_ for 2N per supercell, and 0.714 µ_B_ for 3N per supercell. Surprisingly, only 40% of the magnetic moment came from the Re atoms, and N atoms contributed the other 60% (Figure 8A–C). In addition, the AFM moments mainly stemmed from the 5d orbitals of the Re atoms. Thus, the main contribution of magnetism came from the N atoms, as shown by the electron spin up channel of the Fermi level in Figure 8D. In addition, with an increase in nitrogen content, a strong intermediate gap state appeared close to the Fermi level in Figure 8E–F, which indicated that the electrons could conduct along the Re chain by hopping [18]. In this way, FM and AFM domains were formed, resulting in a strong exchange bias (EB) phenomenon.

Notably, Gao et al. [77] realized an intrinsic RTFM by the adsorption of P onto ReS_2_ nanosheets. Firstly, ReS_2_ powder was synthesized using a hydrothermal method with ammonium rhenate (NH_4_ReO_4_), hydroxylammonium chloride (NH_2_OH·HCl), and thiourea (CH_4_N_2_S) as precursors. Secondly, the obtained powder was placed in a tubular furnace and phosphated with sodium dihydrogen phosphate (NaH_2_PO_2_) in an Ar atmosphere. Furthermore, the adsorption of different concentrations of phosphorus was realized by varying the treatment time and the dose of NaH_2_PO_2_. An RTFM as high as 0.0174 emu/g was experimentally obtained, which was caused by the hybridization of Re *d* and P *p* orbitals in the ReS_2_ supercell with the *T_d_* phase. The control of the phosphating degree could not only realize the modulation of the magnetic coupling strength, but it could also drive the transformation from AFM to FM. In short, RTFM has been achieved by the adsorption of non-metallic atoms.

Fluorination has often been used as a strategy to mediate the desired properties of materials, as shown in Figure 6B. As early as 2009, Zhou et al. [78] found that graphene can be transformed from metallic to semiconducting, from non-magnetic to magnetic, and from direct band gap to indirect band gap by changing the fluorination degree. In addition, fluorination of boron nitride [79] has been found to increase the structural anisotropy and regulate the spin polarization of the system. Experimentally, graphene samples were fluorinated using the CF_4_ radio-frequency plasma technique [80,81,82] or decomposition of xenon difluoride [83,84] at RT. The observed fluorination-regulated magnetism also initiated a theoretical study of fluorine-modulated 2D magnetism. Different from the degree of fluorination on BN [79], which determines whether the system is FM or AFM, the ground state of F-terminated ReS_2_ with *T_d_* phase [21] is AFM. Moreover, its spin configuration depends on the adsorption sites and number of F atoms.

#### 3.2.2. Metal-Doped ReS_2_

In 2014, Peter et al. [70] found that it was easier to incorporate metal atoms into the Re sites. After substitutional doping with metal elements such as Li, Na, V, Cu, Nb, Ta, and Ag, as shown in Figure 9A, the ReS_2_ supercell with *T_d_* phase was still non-magnetic. No matter whether the dopant was residing on the Re site or the S site, doping with Nb and Ta elements could not introduce magnetism in the ReS_2_. However, doping with many other metals, such as Mg, Al, Ti, Cr, Mn, Fe, Co, Zn, Ru, and OS, could introduce magnetism in the ReS_2_, as shown in Figure 9B. Interestingly, when Ti, Mn, and Co elements were substitutionally positioned in the S sites, the magnetism disappeared. The bond length between the transition metal atom and the S atom was also found to modulate the magnetic properties [85]. Doping with two metal atoms has also been studied [74,76,85]. It was found that an increased distance between the metal atoms inhibited magnetism.

### 3.3. Strain Engineering

In 2015, Liu et al. [86] found that a local strain can regulate the optical, electrical, and magnetic properties of single-layer ReSe_2_ (*T_d_* phase) with band gap energy at 1.15 eV. At first, the mechanically exfoliated ReSe_2_ nanosheets were deposited on the pre-stretched elastic substrate. The elastic substrate was then released, and straight-edge wrinkles were introduced into the sheet sample. A local strain was introduced into the sample by creating these wrinkles, by which it was possible to modulate the optical band gap and induce magnetism. As shown in Figure 10, the magnetism in the wrinkled zones could be confirmed using magnetic force microscopy (MFM). Liu et al. carried out density functional theory (DFT) calculations to gain more knowledge about the local strain-regulated magnetism. The results showed that the magnetic moment in the flat area was zero, and the magnetic moment in the wrinkled area had increased to a value close to 3.95 *µ_B_*_._ More specifically, it was found that spin polarization occurred in the wrinkled regions. Furthermore, the effects of uniaxial and biaxial strain on the magnetism were also studied [26]. It was found that compressive strain can annihilate the magnetism of the system and the material can be transformed from half-metal to semiconductor.

However, biaxial strain has never been successfully introduced into Re-based materials, especially in ReS_2_. In 2019, we introduced biaxial strain to the film system by spontaneously forming web buckles [27], as shown in Figure 11. As-grown flat ReS_2_ films were prepared by polymer-assisted deposition [27,43,63,65,66,67,69]. Due to the thermal mismatch between the film and the substrate, the compressive biaxial strain was introduced at the bottom of the ReS_2_ film.

*V_ReS_* and *V_ReS_*_2_ were created after buckling in Figure 10F. The saturation magnetic moment (*M_s_*) at RT was then found to increase from 0.219 emu/g to 0.370 emu/g, as shown in Figure 11A–B. Similarly, the magnetic moments at other temperatures also increased to various degrees, as shown in Figure 11B. In addition, the residual magnetization (*M_r_*) at 5 K increased by a factor of 14, as shown in in Figure 11C. However, the change in coercivity (*H_c_*) was more complex, showing a nonlinear variation with temperature, as depicted in Figure 11D. Moreover, the Curie temperature (T_c_) of the material was greater than 400 K. Interestingly, the in-plane magnetic response was weaker than the out-of-plane magnetic response, which was similar to other typical 2D materials.

In order to clarify the origin of RTFM without and with strain, we also carried out spin density calculations using VASP. In fact, the pristine ReS_2_ crystal with *T_d_* phase was found to be non-magnetic. When Re-related defects were introduced, the system would become magnetic, as shown in in Figure 12A–F. Further, the magnetism clearly changed when a strain was applied to the system. Interestingly, the calculated results showed that the compression strain suppressed the magnetism, and the magnetism became enhanced after a reduction in the compressive strain, as shown in Figure 12G–I. In fact, when the compressive strain decreased from −8% to −5% (*V_Re_*), and from −5% (*V_ReS_*) to −2% (*V_ReS_*_2_), the system could maintain the maximum magnetic moment of 1 µ_B_/supercell, 1 µ_B_/ supercell, and 3 µ_B_/supercell, respectively. However, the magnetic moment remained unchanged with an increase in the tensile strain. Notably, biaxial tensile strain introduced to the system was found to reduce the formation energy of the defects, create more defects, and increase the stability of the defects [87] after buckling. In summary, Re-related defects are not only the origin of RTFM, but they also play a key role in strain-modulated RTFM.

### 3.4. Phase Engineering

ReS_2_ is a direct band gap semiconductor. ReS_2_ is usually in the distorted 1*T* phase (*T_d_* phase), which is different from the 2*H* phase of most transition metal chalcogenides. The low symmetry of the structure leads to its diamagnetism. Yang et al. [16] theoretically predicted a new distorted phase (Tri phase) with tunable magnetism. More specifically, the Re atoms formed a uniform-trimer in the second phase. More importantly, the Tri phase could be achieved using doping [20,88] or intercalating [89] of the *T_c_* phase, and it had bipolar magnetic semiconducting behavior (~1.63 eV) at RT. Moreover, it was predicted that carrier doping could not only realize a transformation from a semiconducting phase to a semi-metallic phase, but also raise the T_c_ to 357K. Furthermore, the overlap of isolated *d* orbitals in the trimer unit forms a direct exchange between *a* (*d_z_*_2_) and *e*_1_ (*d_xy_* and *d_x_*_2_) *d* orbitals in Re atoms, which leads to ferromagnetic coupling. Meanwhile, the superexchange interaction between Re *a* and *e*_1_
*d* orbitals is modulated by the 3*p* S orbitals, forming weak ferromagnetic coupling between the neighboring trimers. In short, the direct ferromagnetic coupling between the Re atoms leads to a huge magnetic anisotropy and a high T_c_. However, the Tri phase has not yet been experimentally obtained.

The migration of out-of-plane electric dipoles was strictly limited by the large potential barrier energy in the *T_d_* phase, which restrained the emergence of ferroelectricity. By introducing a centrosymmetric metal *T_c_* phase (Figure 2B) into the *T_d_* phase (Figure 2C), a new phase *T_t_* could be constructed, which realized an out-of-plane ferroelectricity, as shown in Figure 13A. When the *T_t_* phase was created by the formation of *V_Re_*, a magnetic order in the system was found [17,45,57,90]. Theoretical calculations showed that S atoms at different positions of *V_Re_* will cause obvious changes in the magnetism (Figure 13A). Similar to the ReS_2_ web buckles [27], the out-of-plane FM at RT was about 3.4 times larger than that of the in-plane FM, as shown in Figure 13B. Notably, the mean field approximation showed that T_c_ could be estimated as 704 K, as shown in Figure 13C–D. The observed FM was found to be very close to *V_Re_*.

### 3.5. Domain Engineering

A variety of domain structures are often found in MoS_2_ [91,92,93,94,95,96,97,98,99], WS_2_ [100,101] and ReS_2_ [102] samples that have been prepared by chemical vapor deposition. However, domain engineering is rarely used in exfoliated nanosheets. In 2020, Loh et al. [37] found mirror twin boundaries in pristine ReS_2_ crystals with the *T_d_* phase using linearly polarized OM, ARPRS, and STEM analysis. Pristine ReS_2_ crystals were first obtained by electrochemical exfoliation, but FM at low temperatures was observed in intercalated samples. However, spin-polarized calculations showed that the system was non-magnetic whether there were parallel mirror twin boundaries or not. Moreover, the system showed a magnetism after the introduction of sulfur vacancies. The magnetic moment increased from 0.09 µ_B_/supercell to 1.94 µ_B_/supercell for an applied strain increase of 9%. Interestingly, most of the spins were concentrated on the Re atoms that were close to the grain boundaries. Furthermore, VASP calculations showed that a coexistence lattice strain and *V_s_* at the grain boundaries mainly contributed to FM.

## 4. Conclusions and Outlook

The construction of a relationship between structure, strain, and magnetism has always been a problem to solve. Although biaxial strain-controlled RTFM has been achieved by buckling, the in-situ variation of RTFM with the buckling process is still unclear. In addition, the influence of uniaxial strain on ferromagnetism has not been clarified, although it is generally believed that a biaxial strain should have a stronger impact on the properties than a uniaxial strain. Therefore, it is of great interest to also explore the effect of a uniaxial strain on the RTFM of ReS_2_.

In addition to its potential applications in spintronic materials and devices, ReS_2_ has shown promising potentials in other fields. For instance, ReS_2_ with a stable *T_d_* phase structure has recently been shown to have potential application possibilities in the fields of photocatalysis [86,103,104], hydrogen evolution reactions (HER) [44,104,105,106,107,108,109], and lithium-ion batteries [94,95,96,97,98,99,100,101,102,103]. This is mainly due to the weak interlayer coupling in ReS_2_. Interestingly, an external magnetic field has been used for magnetic catalysts, with the purpose of enhancing the HER and oxygen evolution reaction (OER) activity. Since ReS_2_ is a FM material, it has been assumed that ReS_2_ can be used as an electrocatalyst. In addition, nanoscale magnetic imaging techniques [110], such as nanowire magnetic force microscopy [111], scanning superconducting quantum interference device microscopy (SQUID) [112], and scanning nitrogen-vacancy center microscopy (SNVM) [113,114,115,116], have emerged as important tools in the investigation of 2D materials. These techniques have made it possible to detect magnetism in, e.g., buckled areas. For all the above-mentioned future research directions, a deep understanding of the origin of RTFM and strain-tunable RTFM is necessary, which will further help the development of both basic research and practical applications to flourish.

## Figures and Tables

**Figure 1 nanomaterials-12-03451-f001:**
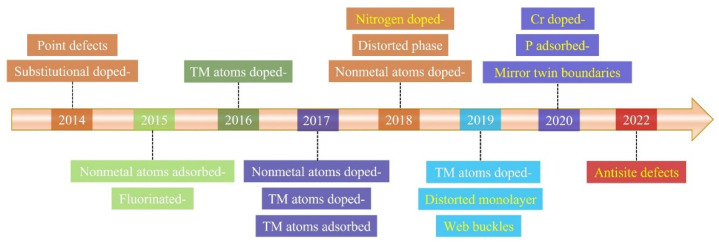
Timeline showing key developments of RTFM in ReS_2_. White font represents the theoretical progress; yellow font represents the experimental progress.

**Figure 2 nanomaterials-12-03451-f002:**
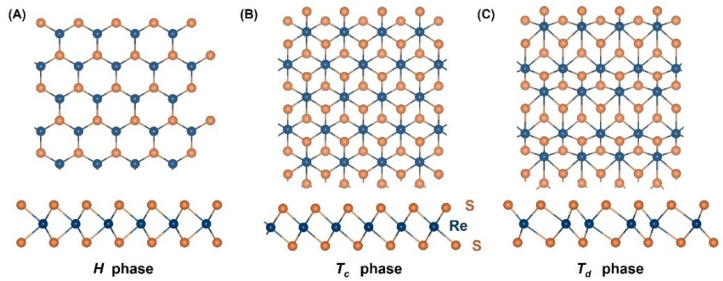
Top and side view of single-layer crystal structures of (**A**) *H* phase, (**B**) *T_c_* phase, and (**C**) *T_d_* phase.

**Figure 3 nanomaterials-12-03451-f003:**
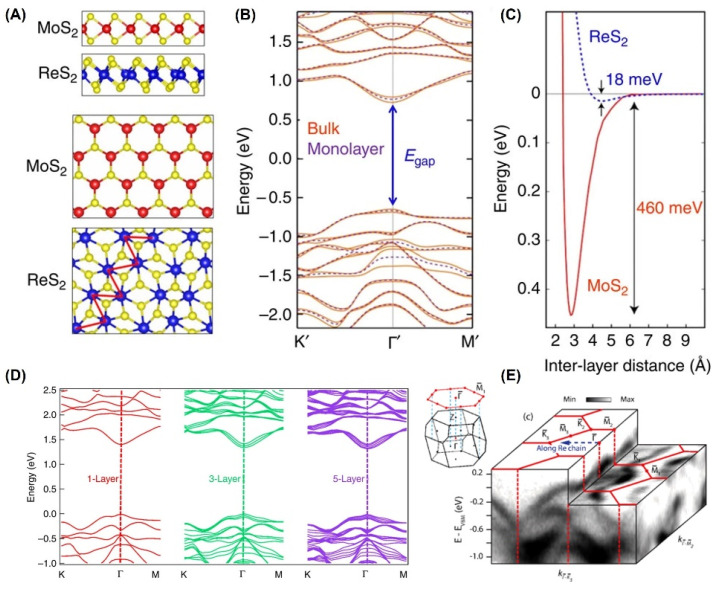
(**A**) Top and side view of single-layer crystal structures of side (top two panels) and top view (bottom two panels) of ReS_2_ with *T_d_* crystal structure compared with the 1*H* structure of conventional monolayer TMDs. The Re atoms dimerize as a result of the Peierls distortion forming a Re chain denoted by the red zigzag line. (**B**) DFT calculated electronic band structure of bulk and monolayer ReS_2_. Both are predicted to be a direct bandgap semiconductor with nearly identical bandgap value at the Γ point. (**C**) The calculated total energy of the system as a function of interlayer separation. The significantly shallower depth of the well in ReS_2_ implies much weaker interlayer coupling energy in ReS_2_ as compared with MoS_2_. (Reprinted figure (**A**–**C**) with permission from [41]. Copyright (2018) by Springer Nature.) (**D**) DFT calculated electronic band structure of monolayer, trilayer, and five-layer ReS_2_ by ab initio calculations indicating band gaps of 1.44, 1.40, and 1.35 eV, respectively. (Reproduced with permission from [51]. Copyright (2018) by Springer Nature.) (**E**) Overview of the valence-band structure as measured by ARPES, showing strong in-plane anisotropy. The surface Brillouin zone is shown as red lines, and the momentum space direction corresponding to the real-space direction along the Re chains is also indicated. The bulk and projected surface Brillouin zones are shown in the inset. (Reprinted figure with permission from [52]. Copyright (2014) by the American Physical Society).

**Figure 4 nanomaterials-12-03451-f004:**
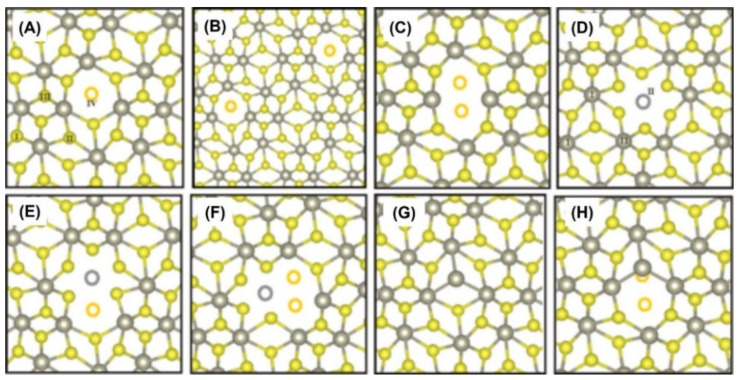
Atomic structure of (**A**) monosulfur vacancy *V_S_*_;_ (**B**) two monosulfur vacancies *V_S+S_*_;_ (**C**) disulfur vacancy *V*_2*S*;_ (**D**) Re vacancy *V_Re_*_;_ (E) ReS vacancy *V_ReS_*_;_ (**F**) ReS_2_ vacancy *V_ReS_*_2;_ (**G**) single S atom substituted by a Re atom *S_S_**_→Re_*; and (**H**) two S atoms substituted by a Re atom *S*_2*S*_*_→Re_*. (Reprinted figure with permission from [57]. Copyright (2014) by the American Physical Society.)

**Figure 5 nanomaterials-12-03451-f005:**
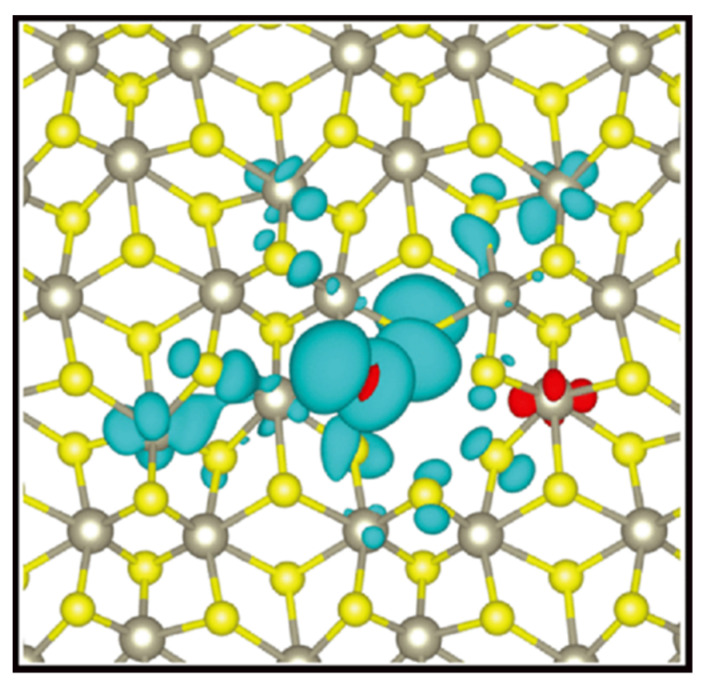
The spin resolved charge density for Re vacancy. Bule (red) colors represent spin up (down) electrons. (Reprinted figure with permission from [57]. Copyright (2014) by the American Physical Society).

**Figure 6 nanomaterials-12-03451-f006:**
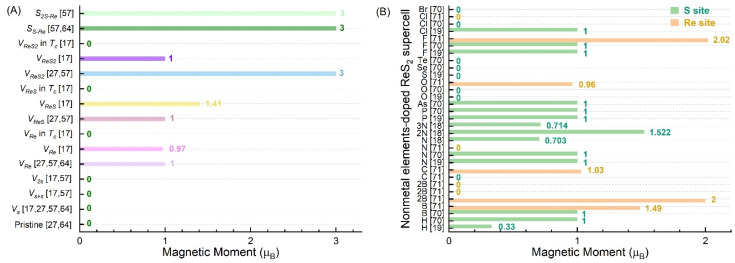
Calculated total magnetic moment in ReS_2_ supercell with different point defects (**A**) and different nonmetallic elements doped (**B**) [17,18,19,27,57,64,70,71]; *T_d_* phase: noncentrosymmetric; *T_c_* phase [17]: centrosymmetric metallic.

**Figure 7 nanomaterials-12-03451-f007:**
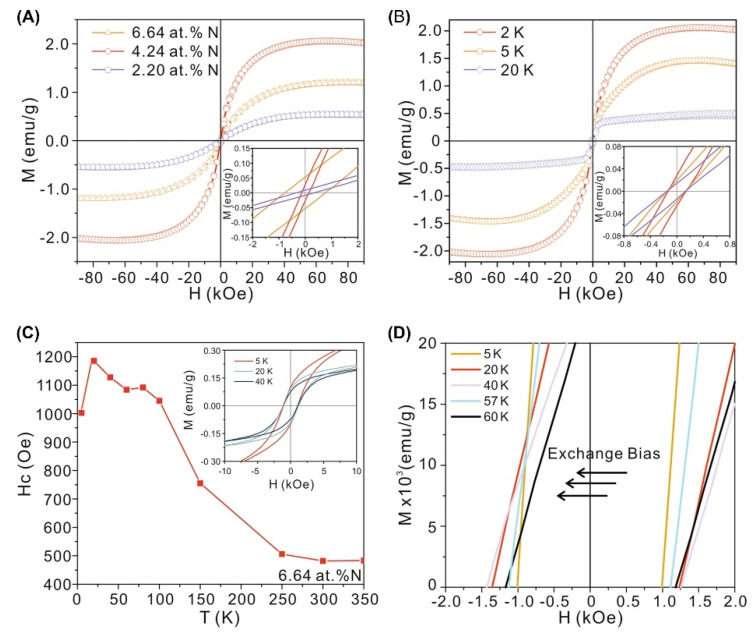
*M*−*H* of ReN_x_ with different nitrogen doping concentrations and detailed analysis with temperature evolution. (**A**) *M*−*H* of ReN_x_ at 2K; the inset shows the non−zero coercivity around the origin. (**B**) *M*−*H* of ReN_4.24_ at low temperatures. (**C**) *H_c_ vs T* of the ReN_6.64_ sample. The inset shows the exchange bias around the Néel point. (**D)** Magnified view of the *M*−*H* of ReN_6.64._ (Reproduced with permission from [18]. Copyright 2018, Springer Nature.)

**Figure 8 nanomaterials-12-03451-f008:**
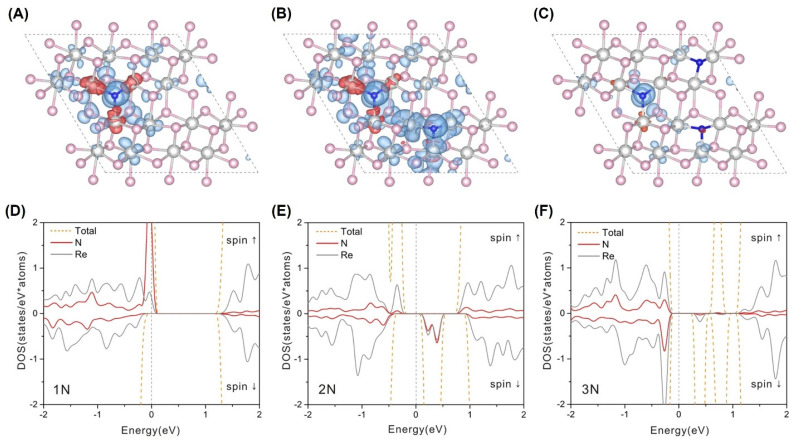
(**A**−**C**) The configuration view of ReN_x_, with positive (negative) spin densities plotted in blue (red) for isosurfaces at 1 × 10^−3^ eÅ^−3^. (**D**−**F**) The corresponding DOS plots of ReS_2_ doped with 1N, 2N, and 3N, respectively. (Reproduced with permission from [18]. Copyright 2018, Springer Nature).

**Figure 9 nanomaterials-12-03451-f009:**
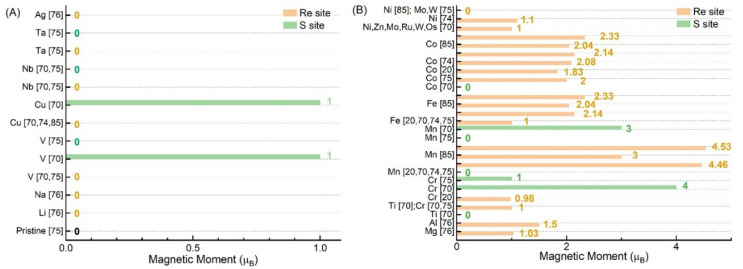
Total magnetic moment of the metal-element doped ReS_2_ [20,70,74,75,76,85]; (**A**) pristine− and doped−ReS_2_ supercell at the Re site were still nonmagnetic. (**B**) doped−ReS_2_ supercell with one or two atoms exhibited rich magnetic behaviors.

**Figure 10 nanomaterials-12-03451-f010:**
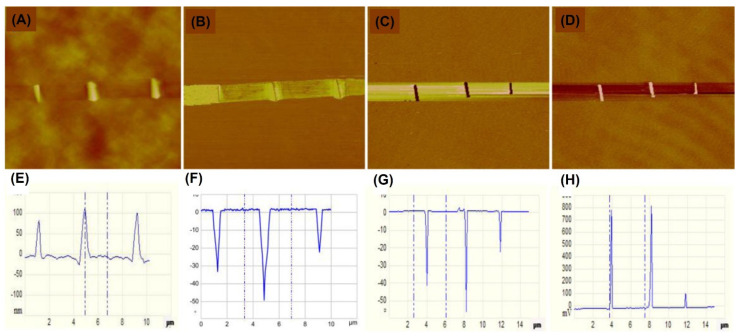
Magnetic force microscopy measurements on ReSe_2_. (**A**) AFM topography, (**B**) phase, (**C**) MFM phase, and (**D**) MFM amplitude images of monolayer ReSe_2_ wrinkled flake on gel−film subs−trate. (**E**−**H**) The corresponding profiles in panels (**A**−**D)**. Reprinted with permission from [86]. Copyright 2015, American Chemical Society.

**Figure 11 nanomaterials-12-03451-f011:**
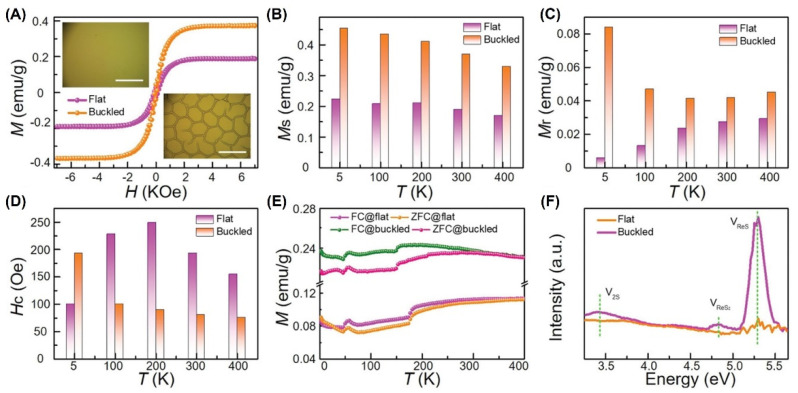
Ferromagnetism of ReS_2_ thin films. (**A**) M−H curves at 300 K. Scale bar: 50 µm. (**B**−**D**) *M_s_*−*T*, *M_r_*−T, and *H_c_*−*T*, respectively. (**E**) FC and ZFC curves from 2 to 400 K. (**F**) FL curves. Reprinted with permission from [27]. Copyright 2019, John Wiley and Sons.

**Figure 12 nanomaterials-12-03451-f012:**
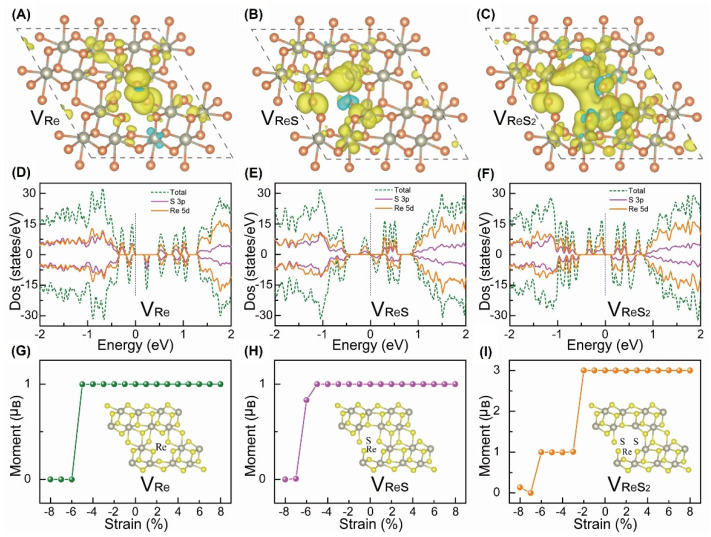
(**A**−**C**) Magnetic density maps of V_Re_, V_ReS_, and V_ReS2_ of ReS_2_ crystals with up (down) spin densities plotted in yellow (blue) for isosurfaces at 0.001 eÅ^−3^, respectively. (**D**−**F**) Orbital projection density of states (PDOS) of V_Re_, V_ReS_, and V_ReS2_ of ReS_2_ crystals, respectively. (**G**−**I**) The variation in magnetic moment of ReS_2_ crystal with V_Re_, V_ReS_, and V_ReS2_ under biaxial compressive and tensile strain. The inset shows the corresponding crystal structure with vacancy defects. Please note that when the strain value is negative, the strain is compression strain; when the strain is positive, it is a tensile strain. Reprinted with permission from [27]. Copyright 2019, John Wiley and Sons.

**Figure 13 nanomaterials-12-03451-f013:**
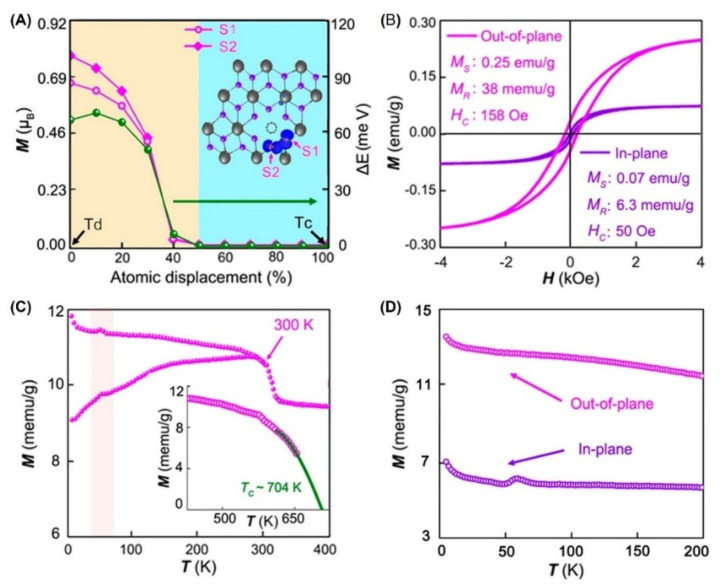
(**A**) Calculated magnetic moment (magenta lines) of atoms S1 and S2 and energy difference between the spin-polarized and spin-unpolarized states (olive line) versus atomic displacement from the *T_d_* to *T_c_* structures (the solid line is a guide for the eye). The inset shows the spin-resolved charge density of the two immediate neighbor atoms S1 and S2 near the Re vacancy (dashed circle). (**B**) Orientation dependence of magnetization at 300 K. (**C**) Temperature dependence of magnetic-field cooling (FC) and zero magnetic-field cooling (ZFC). The extrapolated line at higher temperatures intersects the temperature axis at 704 K, indicating the Curie temperature (inset). (**D**) Temperature dependence of the magnetic susceptibility in the out-of-plane and in-plane directions. Reprinted with permission from [17]. Copyright 2019, American Chemical Society.

## Data Availability

Not applicable.

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
