# Peer review of "Recent Progress in Research on Ferromagnetic Rhenium Disulfide"

_nanomaterials, 2022, doi:10.3390/nano12193451_

Round 1

Reviewer 1 Report

The authors summarized the progress in the research of inducing ferromagnetism in 2D ReS2.
The covered topics include defect introduction, chemical substitution, strain, structural modification by the incorporation of another phase, and the use of twin boundaries, about which they seem to be rather skeptical.

They showed that theoretical calculations have been a good guideline for experimental efforts in this field.
The number of citations are good enough and they seem to have cited the works of different groups fairly.

This review can be a compact introduction to the research of magnetism not only in ReS2, but other 2D materials.

I have no particularly major concern on this paper.
I just have some minor comments.

1. Since this is a review paper and the readership is not limited to the researchers of ReS2, it will be better to explain in the Introduction why ReS2 is particularly important among other TM dichalcogenides.

2. In line 14 on page 3, it goes "The mechanism defect of the modulated magnetism was then revealed." Is there an error in this sentence possibly?

3. In the caption of Fig. 3, (h)  should be capitalized and bold.

4. In the second paragraph of page 4, the words "web buckle" appears. Since this term was new to me, I took a look at the Ref. 27, but here again this word was used without definition. I also looked up "web buckle" at Google Scholar, but all the usages other than by the authors' group were related to architecture, meaning buckling in the web part of some beam structure.  I can imagine the implication of these words, but not for sure. I really appreciate if the authors give a definition here.

5. In the caption of Fig. 5, Date should read Data.

6. In line #4 from the bottom on page 5, the use of "the weakest" seems inappropriate to me since there were elements with no effect. "modest"  will be a better choice.

7. In line 1 on page 7, A RTFM should read An RTFM.

8. In line 6 on page 7, a word "desired" or something similar may well be added before "properties".

9. In line #9 from the bottom on page 8, the hyphen after "temperature" should be omitted.

10. In line #3 from the bottom on page 8, "magnetic" should read "non-magnetic"?

11. In the first paragraph of page 9, it would be kinder to readers to comment that a negative strain value in Fig. 11 (G)-(I) means compressive strain and a positive one tensile.

12. In the caption of Fig. 11, eAngstrom^3 should be eAngstrom^-3.

13. In line 6 on page 10, "a second phase" is better to be "the second phase" since it has been mentioned in the previous paragraph.

14. In the caption of Fig. 12, unpolaize should be unpolarized and oliveone needs a blank space between olive and one.

Author Response

Response to Reviewer 1

The authors summarized the progress in the research of inducing ferromagnetism in 2D ReS2. The covered topics include defect introduction, chemical substitution, strain, structural modification by the incorporation of another phase, and the use of twin boundaries, about which they seem to be rather skeptical.

They showed that theoretical calculations have been a good guideline for experimental efforts in this field.

The number of citations are good enough and they seem to have cited the works of different groups fairly. This review can be a compact introduction to the research of magnetism not only in ReS2, but other 2D materials.

I have no particularly major concern on this paper. I just have some minor comments.

[Comment]:

Point1. Since this is a review paper and the readership is not limited to the researchers of ReS2, it will be better to explain in the Introduction why ReS2 is particularly important among other TM dichalcogenides.

[Response]:

Thank you very much for taking time to review our manuscript and for the positive comments on our manuscript. In response to your point 1, we have added the following sentences on page 1-4 in the revised manuscript.

“Unlike other hexagonal (H or 2H) transition metal chalcogenides (TMDs) with high symmetry, the unique distorted (Td) structure of rhenium disulfide (ReS2) has a low in-plane symmetry [41], which endows it with strong in-plane anisotropic properties [42-44]. It is worth noting that a series of progress has also been made in the RTFM of ReS2.” on page 1-2.

“As shown in Figure 3(A), the crystal structure of ReS2 with Td phase [41] is obviously different from that of MoS2 with H phase [49, 50]. Density functional theory (DFT) calculations show that bulk (1.35 eV) and monolayer (1.43 eV) ReS2 have similar band structures, both of which are direct band gap semiconductors, but their band gaps are slightly different (only 80 mV difference) in Figure 3(B).

Figure 3. (A) Top and side view of single-layer crystal structures of side (top two panels) and top view (bottom two panels) of ReS2 with Td crystal structure compared with the 1H structure of conventional monolayer TMDs. The Re atoms dimerize as a result of the Peierls distortion forming a Re chain denoted by the red zigzag line. (B) DFT calculated electronic band structure of bulk and monolayer ReS2. Both are predicted to be a direct bandgap semiconductor with nearly identical bandgap value at the Γ point. (C) The calculated total energy of the system as a function of interlayer separation. The significantly shallower depth of the well in ReS2 implies much weaker interlayer coupling energy in ReS2 as compared with MoS2. (Reprinted figure (A-C) with permission from [41]. Copyright (2018) by Springer Nature). (D) DFT calculated electronic band structure of monolayer, trilayer and five-layer ReS2 by ab initio calculations indicating band gaps of 1.44, 1.40 and 1.35 eV, respectively. (Reproduced with permission from [51]. Copyright (2018) by Springer Nature). (E) Overview of the valence-band structure as measured by ARPES, showing strong in-plane anisotropy. The surface Brillouin zone is shown as red lines, and the momentum space direction corresponding to the real-space direction along the Re chains is also indicated. The bulk and projected surface Brillouin zones are shown in the inset. (Reprinted figure with permission from [52] Copyright (2014) by the American Physical Society).

Actually, the adjacent layers in ReS2 are only weakly coupled (~18 meV) in Figure 3(C), while those in MoS2 are coupled with much higher energy (~460 meV). Interestingly, when the thickness of ReS2 is reduced down to a single layer [51], its electronic band structure does not exhibit a transition from an indirect to a direct bandgap, which is different from that of MoS2.” on page 3-4.

[Comment]:
Point 2: In line 14 on page 3, it goes "The mechanism defect of the modulated magnetism was then revealed." Is there an error in this sentence possibly?

[Response]:

As the reviewer pointed out, we have revised the relevant sentences on page 4.

“The mechanism of defect-mediated magnetism was then revealed.”

[Comment]:

Point 3: In the caption of Fig. 3, (h) should be capitalized and bold.

[Response]:

Thank you very much for your comment. According to your suggestion, we have revised the caption of Fig. 4(H) on page 4.

[Comment]:

Point 4: In the second paragraph of page 4, the words "web buckle" appears. Since this term was new to me, I took a look at the Ref. 27, but here again this word was used without definition. I also looked up "web buckle" at Google Scholar, but all the usages other than by the authors' group were related to architecture, meaning buckling in the web part of some beam structure. I can imagine the implication of these words, but not for sure. I really appreciate if the authors give a definition here.

[Response]:

We thank the reviewer for this valuable suggestion. According to your suggestion, we have added the two references related to web buckles as Refs. 68 and 69, and added given a definition on page 5 of the revised manuscript.

“Buckles exist in multiple directions in the plane, and cross each other to form some web patterns.”

[Comment]:

Point 5: In the caption of Fig. 5, Date should read Data.

[Response]:

Thank you very much for your comment. According to your suggestion, we have revised the caption of Fig.6 on page 6 of the revised manuscript.

[Comment]:

Point 6: In line #4 from the bottom on page 5, the use of "the weakest" seems inappropriate to me since there were elements with no effect. "modest" will be a better choice.

[Response]:

Thank you very much for your comment. According to your suggestion, we have substituted “modest” for “weakest” on page 6.

[Comment]:

Point 7: In line 1 on page 7, A RTFM should read An RTFM.

[Response]:

Thank you very much for your comment. According to your suggestion, we have modified the sentence on page 8 of the revised manuscript.

[Comment]:

Point 8: In line 6 on page 7, a word "desired" or something similar may well be added before "properties".

[Response]:

Thank you very much for your helpful suggestion. Following your suggestion, we have added the word “desired” before “properties” sentence on page 8.

[Comment]:

Point 9: In line #9 from the bottom on page 8, the hyphen after "temperature" should be omitted.

[Response]:

Thank you very much for your helpful suggestion. Following your suggestion, we have removed the hyphen on page 9.

[Comment]:

Point 10: In line #3 from the bottom on page 8, "magnetic" should read "non-magnetic"?

[Response]:

Thank you very much for pointing out this mistake. Following your suggestion, we have revised the sentence on page 9 in our manuscript.

[Comment]:

Point 11: In the first paragraph of page 9, it would be kinder to readers to comment that a negative strain value in Fig. 11 (G)-(I) means compressive strain and a positive one tensile.

[Response]:

Thank you very much for your helpful suggestion. Following your suggestion, we have revised the sentence in the first paragraph of page 10, and added the relevant description in the caption of Fig. 12 on page 10.

“In fact, when the compressive strain decreased from -8% to -5% (VRe), -5% (VReS) and -2% (VReS2), the system could maintain the maximum magnetic moment of 1 µB/supercell, 1 µB/supercell and 3 µB/supercell, respectively.”

“Please note that when the strain value is negative, the strain is compression strain; When the strain is positive, it is a tensile strain.”

[Comment]:

Point 12: In the caption of Fig. 11, eAngstrom^3 should be eAngstrom^-3.

[Response]:

Thank you very much for pointing out this mistake. Following your suggestion, we have revised the sentence in the caption on page 10.

[Comment]:

Point 13: In line 6 on page 10, "a second phase" is better to be "the second phase" since it has been mentioned in the previous paragraph.

[Response]:

Thank you very much for your helpful suggestion. Following your suggestion, we have revised the sentence on page 11.

[Comment]:

Point 14: In the caption of Fig. 12, unpolaize should be unpolarized and oliveone needs a blank space between olive and one.

[Response]:

Thank you very much for pointing out this mistake. We have corrected it in the caption of Fig. 13, and added a blank space between olive and one on page 11.

Reviewer 2 Report

This review is well written and regards an exciting topic as the ferromagnetism in 2D systems. However, many improvements are necessary before publication in my opinion.

1) In the first part of the introduction, the authors could also cite the attempt to make ferromagnetic or ferrimagnetic systems as MPS3 that are antiferromagnetic when undoped. For instance, this was done in these two papers:

A) The Journal of Physical Chemistry C 126 (15), 6791-6802 (2022)

B) Rabindra Basnet et al 2022 J. Phys.: Condens. Matter 34 434002

2)  For all phases presented in Figure 2, the authors need to provide a band structure, or at list they should clarify if they are all insulating and with which gap (size and direct/indirect)

3) Along all the paper, the authors should better specify if the system is metallic or insulator in the several cases they consider. If possible, they should also specify better the crystal structure of ReS2 in the several cases.

4) More attention should be dedicated to the magnetic mechanism for the several cases investigate. Do you have double exchange, superexchange ?? what is the mechanism for the ferromagnetic? This should be mentioned in abstract or conclusion. If this is completely unknow, this should be indicated as a future research direction.

OTHER MINOR CORRECTIONS

1) from stylistic point of view, I would suggest to put the title on two lines as "Research progress in Research about" and "Ferromagnetic ...."

2) This sentence should be rephrased: "Notably, only a few ReS2 materials exhibited long-range ferromagnetic ordering at room temperature (RT) in laboratory"

After these corrections, I will be happy to reconsider this paper for publication.

Author Response

Response to Reviewer 2

This review is well written and regards an exciting topic as the ferromagnetism in 2D systems. However, many improvements are necessary before publication in my opinion.

[Comment]:

  1. In the first part of the introduction, the authors could also cite the attempt to make ferromagnetic or ferrimagnetic systems as MPS3 that are antiferromagnetic when undoped. For instance, this was done in these two papers:
  2. A) The Journal of Physical Chemistry C 126 (15), 6791-6802 (2022).
  3. B) Rabindra Basnet et al 2022 J. Phys.: Condens. Matter 34 434002.

[Response]:

Thank you very much for taking time to review our manuscript and for the positive comments on our manuscript.

Following your suggestion, we have cited these two papers as Refs. 38 and 39, and added a brief review on these two papers on page 1 in the revised manuscript:

“Interestingly, transition metal phosphorus trichalcogenides (MPX3, where M stands for a transition metal atom and X = S and Se) have also attracted much attentions [38, 39]. In particular, the pristine MPS3 exhibited antiferromagnetic (AFM). When the "M" atom is replaced by other transition metal atoms, it may drive the transition from AFM to FM.” on page 1.

[Comment]:

  1. For all phases presented in Figure 2, the authors need to provide a band structure, or at list they should clarify if they are all insulating and with which gap (size and direct/indirect).

[Response]:

Thank you very much for taking time to review our manuscript and the positive comments on our manuscript. According to your suggestion, we have added the new section (2.2 band structure) and figure (as Figure 3) on page 3-4 in the revised manuscript, as follow:

2.2. Band Structure

Bulk ReS2 is a direct band gap semiconductor with a layered structure, showing novel anisotropic properties [41, 48]. As shown in Figure 3(A), the crystal structure of ReS2 with Td phase [41] is obviously different from that of MoS2 with H phase [49, 50]. Density functional theory (DFT) calculations show that bulk (1.35 eV) and monolayer (1.43 eV) ReS2 have similar band structures, both of which are direct band gap semi-conductors, but their band gaps are slightly different (only 80 mV difference) in Figure 3(B).

Figure 3. (A) Top and side view of single-layer crystal structures of Side (top two panels) and top view (bottom two panels) of ReS2 with Td crystal structure compared with the 1H structure of conventional monolayer TMDs. The Re atoms dimerize as a result of the Peierls distortion forming a Re chain denoted by the red zigzag line. (B) DFT calculated electronic band structure of bulk and monolayer ReS2. Both are predicted to be a direct bandgap semiconductor with nearly identical bandgap value at the Γ point. (C) The calculated total energy of the system as a function of interlayer separation. The significantly shallower depth of the well in ReS2 implies much weaker interlayer coupling energy in ReS2 as compared with MoS2. (Reprinted figure (A-C) with permission from [41]. Copyright (2018) by Springer Nature). (D) DFT calculated electronic band structure of monolayer, trilayer and five-layer ReS2 by ab initio calculations indicating band gaps of 1.44, 1.40 and 1.35 eV, respectively. (Reproduced with permission from [51]. Copyright (2018) by Springer Nature). (E) Overview of the valence-band structure as measured by ARPES, showing strong in-plane anisotropy. The surface Brillouin zone is shown as red lines, and the momentum space direction corresponding to the real-space direction along the Re chains is also indicated. The bulk and projected surface Brillouin zones are shown in the inset. (Reprinted figure with permission from [52] Copyright (2014) by the American Physical Society).

Actually, the adjacent layers in ReS2 are only weakly coupled (~18 meV) in Figure 3(C), while those in MoS2 are coupled with much higher energy (~460 meV). Interestingly, when the thickness of ReS2 is reduced down to a single layer [51], its electronic band structure does not exhibit a transition from an indirect to a direct bandgap, which is different from that of MoS2. The bandgaps of bulk, trilayer and monolayer ReS2 are 1.35 eV, 1.40 eV, and 1.44 eV, respectively, as shown in Figure 3(D). In experiment, the electrical band structure of rhenium disulfide can be accurately described by angle-resolved photoemission spectroscopy (ARPES) measurements [52]. Although the surface Brillouin zone of ReS2 is almost hexagonal, its electronic structure shows significant in-plane anisotropy, resulting in unique anisotropic optical and electrical properties.

[Comment]:

  1. Along all the paper, the authors should better specify if the system is metallic or insulator in the several cases they consider. If possible, they should also specify better the crystal structure of ReS2 in the several cases.

[Response]:

Thank you very much for your helpful suggestion. Following your suggestion, we have specified the crystal structure and band energy structure of ReS2 in the several cases.

“As shown in Figure 3(A), the crystal structure of ReS2 with Td phase [41] is obviously different from that of MoS2 with H phase [49, 50]. Density functional theory (DFT) calculations show that bulk (1.35 eV) and monolayer (1.43 eV) ReS2 have similar band structures, both of which are direct band gap semiconductors, but their band gaps are slightly different (only 80 mV difference) in Figure 3(B).” on page 3.

“Optimized atomic structures of distorted 1T-ReS2 monolayer were then created, as shown in Figure 4. However, the introduction of defects in these optimized structures was not found to change the semiconductor properties or drive any phase transition.” on page 4.

“Actually, the defects in ReS2 nanosheets were formed by the Re atoms occupying the positions of the S atoms, and new VRe defects were simultaneously introduced (Figure 6).” on page 5.

“In order to explain the correlation between doping concentration and magnetism for the ReS2 supercell with Td phase, the magnetic moment and charge distribution were calculated by using VASP in Figure 8.” on page 7.

“An RTFM as high as 0.0174 emu/g was experimentally obtained, which was caused by the hybridization of Re d and P p orbitals in ReS2 supercell with Td phase.”; “In 2014, Peter et al. [70] found that it was easier to incorporate metal atoms into the Re sites. After substitutional doping with metal elements such as Li, Na, V, Cu, Nb, Ta and Ag in Figure 9(A), the ReS2 supercell with Td phase was still non-magnetic.”; “Different from the degree of fluorination on BN [79], which determines whether the system is FM or AFM, the ground state of F-terminated ReS2 with Td phase [21] is AFM.”; “In 2015, Liu et al. [86] found that local strain can regulate the optical, electrical and magnetic properties of single-layer ReSe2 (Td phase) with band gap energy at 1.15 eV).” on page 8.

“In fact, the pristine ReS2 crystal with Td phase was found to be non-magnetic.” on page 9.

“Yang et al. [16] theoretically predicted on a new distorted phase (Tri phase) with tunable magnetism. More specifically, the Re atoms formed a uniform-trimer in the second phase. More importantly, the Tri phase could be achieved by using doping [20, 88] or intercalating [89] of the Tc phase, and was bipolar magnetic semiconducting (~1.63 eV) at RT. Moreover, it was predicted that carrier doping could not only realize a transformation from semiconducting phase to a semi-metallic phase, but also raise the Tc to 357K. In short, the direct ferromagnetic coupling between the Re atoms lead to a huge magnetic anisotropy and a high Tc. However, the Tri phase has not yet been experimentally obtained.”; “The migration of out-of-plane electric dipoles was strictly limited by the large potential barrier energy in the Td phase, which restrained the emergence of ferroelectricity. By introducing a centrosymmetric metal Tc phase (Figure 2B) into the Td phase (Figure 2C), a new phase Tt could be constructed, which realized an out-of-plane ferroelectricity in Figure 13(A). When the Tt phase was created by the formation of VRe, a magnetic order in the system was found [17, 45, 57, 90].” on page 11.

“In 2020, Loh et al. [37] found mirror twin boundaries in pristine ReS2 crystals with Td phase by using linearly polarized OM, ARPRS and STEM analysis.” on page 12.

[Comment]:

  1. More attention should be dedicated to the magnetic mechanism for the several cases investigate. Do you have double exchange, superexchange ?? what is the mechanism for the ferromagnetic? This should be mentioned in abstract or conclusion. If this is completely unknow, this should be indicated as a future research direction.

[Response]:

We thank the reviewer for this valuable suggestion. Following your suggestion, we have added the sentence in the revised manuscript on page 1 and 11, in response to this comment:

“In addition, the superexchange interaction may cause weak ferromagnetic coupling between the neighboring trimers.” on page 1 in abstract.

“Furthermore, the overlap of isolated d orbitals in the trimer unit forms the direct exchange between a (dz2) and e1 (dxy and dx2) d orbitals in Re atoms, which leads to ferromagnetic coupling. Meanwhile, the superexchange interaction between Re a and e1 d orbitals is modulated by the 3p S orbitals, forming weak ferromagnetic coupling between the neighboring trimers.” on page 11.

[Comment]:

  1. from stylistic point of view, I would suggest to put the title on two lines as "Research progress in Research about" and "Ferromagnetic ...."

[Response]:

We thank the reviewer for this valuable suggestion. According to your suggestion, we have modified the title on page 1 in our manuscript:

“Recent Progress in Research about Ferromagnetic Rhenium Disulfide”

[Comment]:

  1. This sentence should be rephrased: "Notably, only a few ReS2 materials exhibited long-range ferromagnetic ordering at room temperature (RT) in laboratory"

[Response]:

We thank the reviewer for this valuable suggestion. According to your suggestion, we have revised the sentence:

“Experimentally, only a few ReS2 materials exhibited room-temperature long-range ferromagnetic order.” on page 1.

Round 2

Reviewer 2 Report

The authors implemented my suggestions. I strongly recommend the paper for publication in the present form.